# Isotopic and Chemical Tracing for Residence Time and Recharge Mechanisms of Groundwater under Semi-Arid Climate: Case from Rif Mountains (Northern Morocco)

Mohammed Hssaisoune [1,2,*], Lhoussaine Bouchaou [1,3], Mohamed Qurtobi [4], Hamid Marah [4], Mohamed Beraaouz [1] and Jamal Stitou El Messari [5]

[1] Applied Geology and Geo-Environment Laboratory, Faculty of Sciences, Ibn Zohr University, Agadir 80000, Morocco; l.bouchaou@uiz.ac.ma (L.B.); mberaaouz@yahoo.com (M.B.)

[2] Faculty of Applied Sciences, Ibn Zohr University, Ait Melloul 86150, Morocco

[3] International Water Research Institute (IWRI), Mohammed VI Polytechnic University (UM6P), Ben Guerir 43150, Morocco

[4] Unité Eau et Climat, Division des Applications aux Sciences de la Terre et de l'Environnement, CNESTEN, Rabat 10000, Morocco; qurtobi@gmail.com (M.Q.); marah@cnesten.org.ma (H.M.)

[5] Faculty of Sciences, Abdelmalek Essaadi University, Tetouan 93002, Morocco; sjamal@fst.ac.ma

[*] Correspondence: m.hssaisoune@uiz.ac.ma

**Abstract:** Karstic aquifers play an important role for drinking and irrigation supply in Morocco. However, in some areas, a deeper understanding is needed in order to improve their sustainable management under global changes. Our study, based on chemical and isotopic investigation of 67 groundwater samples from the karst aquifer in the Rif Mountains, provides crucial information about the principal factors and processes influencing groundwater recharge and residence time. The $\delta^{18}O$ and $\delta^2H$ isotopic values indicate that the recharge is derived from meteoric water at high, intermediate, and low elevations for Lakraa Mountain, North of Lao River, and Haouz and Dersa Mountain aquifers, respectively. All samples show an isotopic signature from Atlantic Ocean except for those from the Lakraa Mountain aquifer, which shows Mediterranean Sea influence. Groundwater age determined by radiocarbon dating using the IAEA model indicates that the ages range from modern to 1460 years. This short residence time is consistent with the detectable tritium values (>2.7 TU) measured in groundwater. These values are similar to those of precipitation at the nearest GNIP stations of Gibraltar and Fez-Saiss, situated around 100 km north and 250 km south of the study area, respectively. This evidence indicates that groundwater in the Rif Mountains contains modern recharge (<60 years), testifying to significant renewability and the vulnerability of the hydrological system to climate variability and human activities. The results also indicate the efficiency of isotopic tracing in mountainous springs and would be helpful to decision makers for water in this karstic zone.

**Keywords:** karst aquifer; isotopes; residence time; Rif Mountains; Morocco

## 1. Introduction

Karstic aquifers comprise a few of the most productive aquifer and contribute considerably to water supplies in many regions of the world and supply between 20 to 25% of the global population [1]. In the western Mediterranean region, with semi-arid conditions, these aquifers are especially vulnerable to global climate changes [2] and can partly supply the current demand in many countries [3].

In Morocco, carbonate aquifers, especially those in Mountain's area, constitute the most important groundwater resources meeting current water needs of several local populations [4–10]. The populations use the water coming out from the springs for various uses without being very interested in the functioning of the systems from which the water come out. Currently, in Morocco, under the constraint of global climate changes in particular the increase in population, the decrease in precipitation, and the overexploitation of the water



resource at the porous media aquifers in the plains, many shortages in water resources were observed in the country during the last decades [11–13]. The development of the water resources of the mountain aquifers, in addition to non-conventional water (e.g., desalinated seawater, domestic and industrial wastewater, rainwater harvesting, etc.) may constitute for a crucial water supply. In the north of the country, the limestone ridge called "Calcareous Dorsal" of Rif Mountains contains one of the most important productive aquifers drained by numerous springs with large discharges of water and constitute the main resource for drinking and irrigation for some cities and rural communities. These springs, which play significant economic and social roles on hydrological and ecological levels, are so far insufficiently investigated due to their complex context. Sustainable groundwater resource management for these complex aquifers requires a thorough scientific understanding of recharge processes, the mean residence times (MRT) and, consequently, the renewability and vulnerability of the system under human activities and climate changes. The reasons for their high vulnerability include thin soil cover, the flow concentration in the epikarst zone, and concentrated infiltration through karstic porosities. Moreover, the residence time in karstic aquifers is much shorter compared to non-karstic ones.

Chemical and isotopes tracers are among other tools capable of storing information about water–rock interaction, recharge conditions, and the apparent water ages in groundwater systems [14]. Chemical and isotopic tracers have helped characterize many semi-arid aquifers in mountainous areas, where understanding is otherwise limited by unsystematic monitoring and the lack of the necessary equipment [15,16]. The obtained information can be used for various issues such as the determination of recharge areas and the origin of groundwater and widely used for studying the groundwater recharge, migration pathways, and mixing of waters from different sources [14,17–20]. Henceforth, this study based on a combined geochemical and isotopic investigation of 67 groundwater samples from the karst aquifer system of Rif Mountain provide key information for studying the uninvestigated springs such as recharge processes and residence time.

The main objective of this study is to assess the MRT of groundwater coming out from the springs and to provide an idea about their recharge areas. The expected result is to provide more visibility of functioning of investigated karst systems through qualitative information gained from water spring's analysis.

The results of this work will help decision makers and policies to identify additional possible water resources in the region for the potential supply and to manage the water resources in the area marked by the recurrent risk and sensitivity to climate change and anthropogenic impacts.

## 2. Materials and Methods

### 2.1. Study Area

#### 2.1.1. Geographical and Climatic Setting

The Rif area located in the north part of Morocco is bordered by the Mediterranean Sea to the north, the Atlantic Ocean to the west, the Middle Atlas Mountains to the south and to the east (Figure 1a). The climate is influenced by air masses coming from the Atlantic Ocean and Mediterranean Sea circulations (Figure 1b).

According to Salhi et al. [21], the study area is characterized by tempered Mediterranean climate on the coastal area, which is nuanced by altitude and continentality characters, with abundant snow precipitations on the interior reliefs. Average rainfall ranges from 800 to 1400 mm can reach certainly 2000 mm including snow melt on the highest summits (Figure 1c, e and Figure 2). The spatial distribution shows an increase in rainfall from the eastern to western areas (Toreta to Smir station) and from the coastal plains (Toreta, Ben Karrich and Smir stations) to the high mountains (Bab Taza station).

This rate of precipitations in the area is probably due to its geographical position in the northern part of Morocco which characterized by high altitude and double maritime impact (Atlantic and Mediterranean Sea). The Rif area is mainly characterized by humid periods alternates with short dry periods [21].

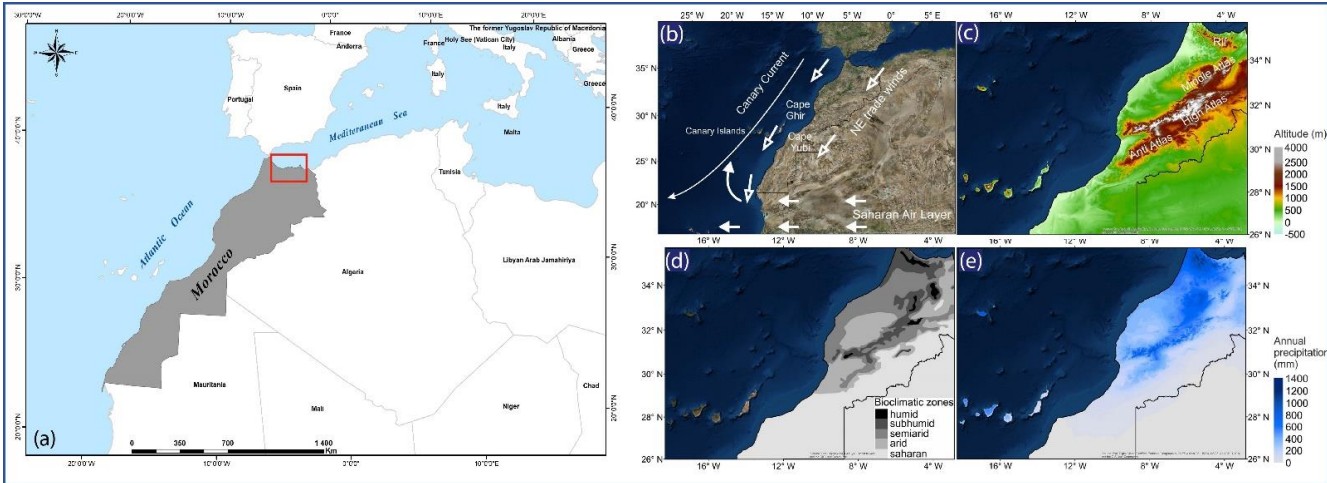

**Figure 1.** (**a**) Geographic location of the studied area (red rectangle) at Mediterranean scale; (**b**) main atmospheric systems (NE trade winds and Saharan Air Layer) and the Canary Current; (**c**) regional topography including the Rif, Middle Atlas, High Atlas, and Anti Atlas Mountain ranges; (**d**) bioclimatic zones; and (**e**) annual precipitation (adapted from [11]).

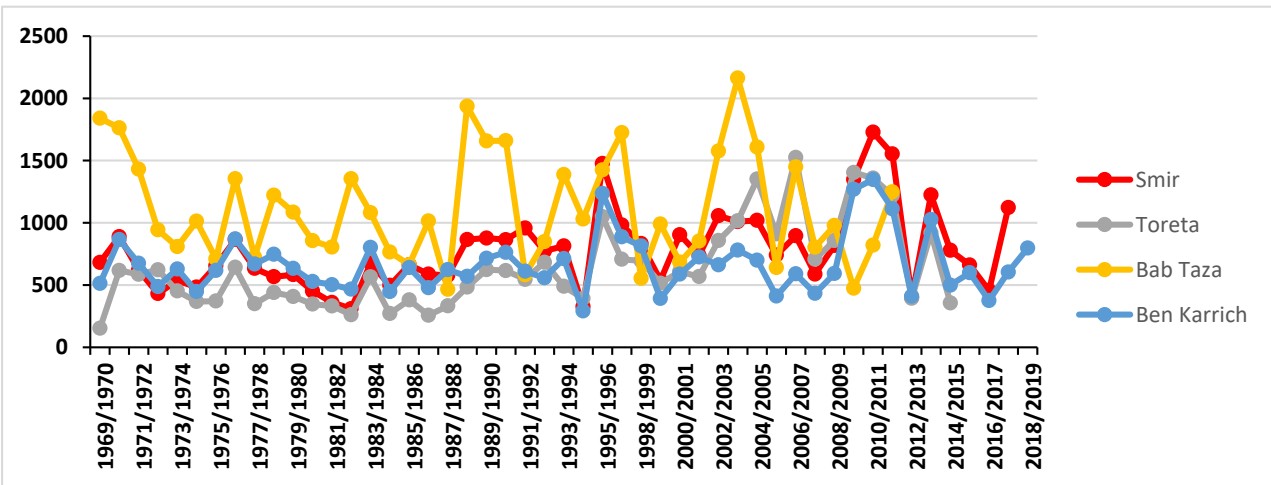

**Figure 2.** Annual variation of precipitation in five main meteorological stations in the Rif region (from 1969 to 2019).

In the Mediterranean humid and sub-humid bioclimatic zone, including Rif Mountains (Figure 1d), the vegetation is rather dense and the Rif shelters a variety of fragmentary forest ecosystems [22]. At first sight, the area appears to be covered by dense vegetation mainly consisting of Atlas Cedar trees (*Cedrus atlantica*) and Moroccan fir trees (*Abies marocana*) that form impressive mountain forests on the limestone reliefs, including evergreen oak and cork oak trees (*Quercus ilex* and *Quercus suber*).

According to several authors [23,24], a well-dated pollen record covering the last 5000 years show clearly the ancient origin of much of the present-day vegetation structures of the study area. The past climate characteristics can influence the dissolution of carbonate formations and consequently the evolution of the karst channels where water passes through since the input area (recharge) to the springs. These phenomena are well known in karst areas [25].

### 2.1.2. Geological and Hydrogeological Setting

The study area is localized in the Internal Zone of the Rif Chain which is characterized by three superimposed tectonic complexes: the "Sebtides," the "Ghomarides" and the

"Calcareous Dorsal" (Figures 3 and 4). The lower Sebtides unit is formed by a deep infra-continental peridotite and overlying polymetamorphic crystalline units and is made of superposed sheets of granulites, gneisses, and micaschists; the upper Sebtides units are mainly Permian metarenites and metapelites with their Triassic Verrucanored beds [26–30].

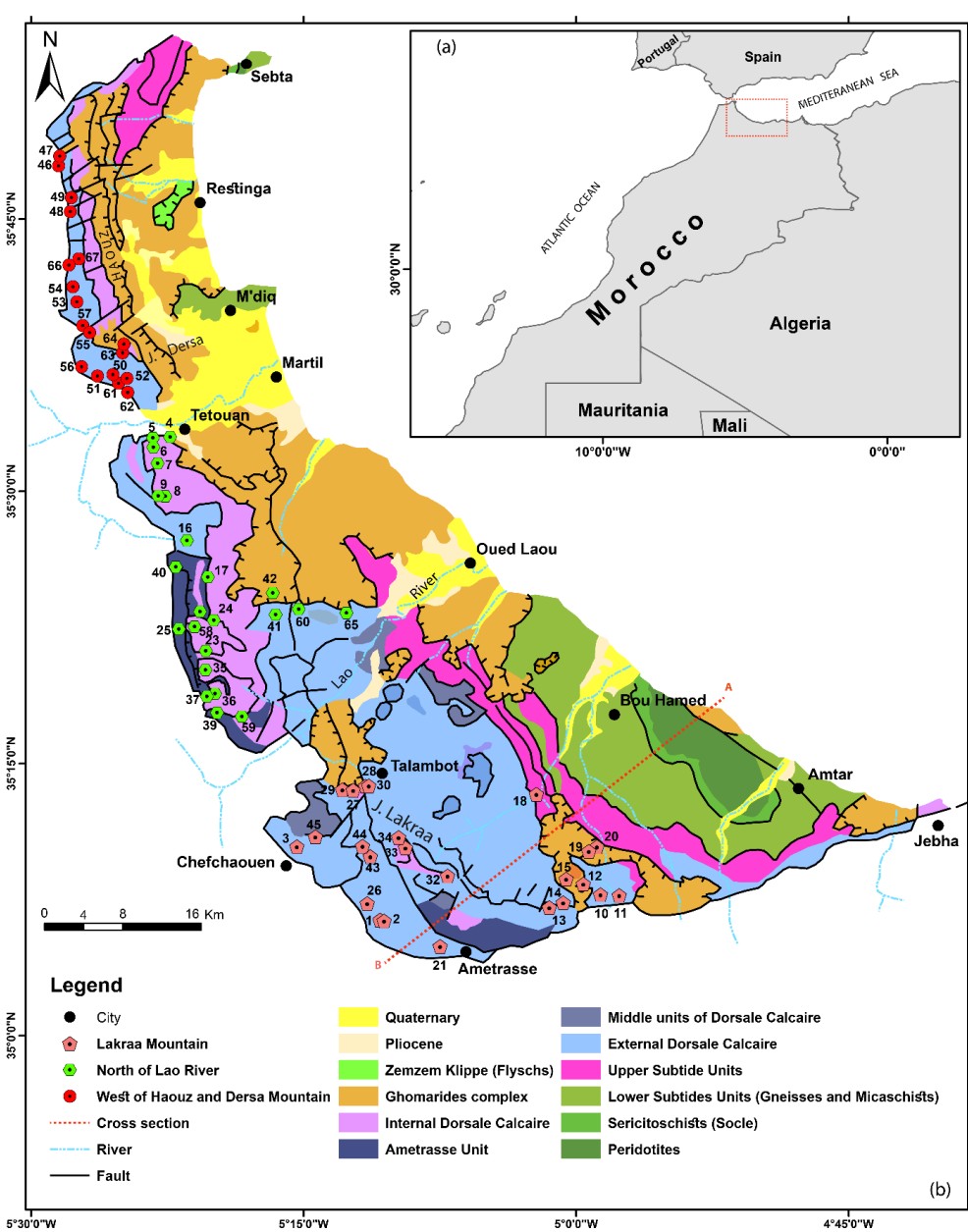

**Figure 3.** (**a**) Location of the study area (dashed rectangle) at the Mediterranean region; (**b**) geological map of the Calcareous Dorsal of Rif adopted from geological map of Morocco (Rif map 1:500,000). The sampled springs from different area are plotted.

The Ghomaride Complex is organized into four Paleozoic nappes with their overlying Triassic–Cenozoic sedimentary cover [26,29]. Finally, the "Calcareous Dorsal" complex, upward to the Ghomaride unit, consists of Triassic-lower Jurassic carbonates evolving upward into Cretaceous slope and basin deposits [26,29,30]. The Calcareous Dorsal Complex is topped by an Aquitanian to Late Burdigalian "syn-orogenic flysch-type" deposit [26].

The study area covers a surface of nearly 825 km$^2$ and corresponds to the limestone chain of the Rif. It extends from the region of Sebta in the north to that of Al Hoceima in the east. It is subdivided into three main units:

- The Haouz to the north of Tetouan, the limestone ridge between Tetouan and Jebha, and the chain of Bokoya to the west of the city of Al Hoceima;
- The Haouz chain extends between Sebta and Tetouan in the form of a narrow strip of about thirty kilometers long. It covers an area of 134 km$^2$. From Tleta Taghramt, this chain breaks up into several small massifs, of which Jbel Moussa is the last link in the north;
- The limestone ridge south of Tetouan: extends from Tetouan to Jebha; it consists mainly of a thick series of limestone and dolomitic Trias-Lias strongly tectonized, karstified and cracked.

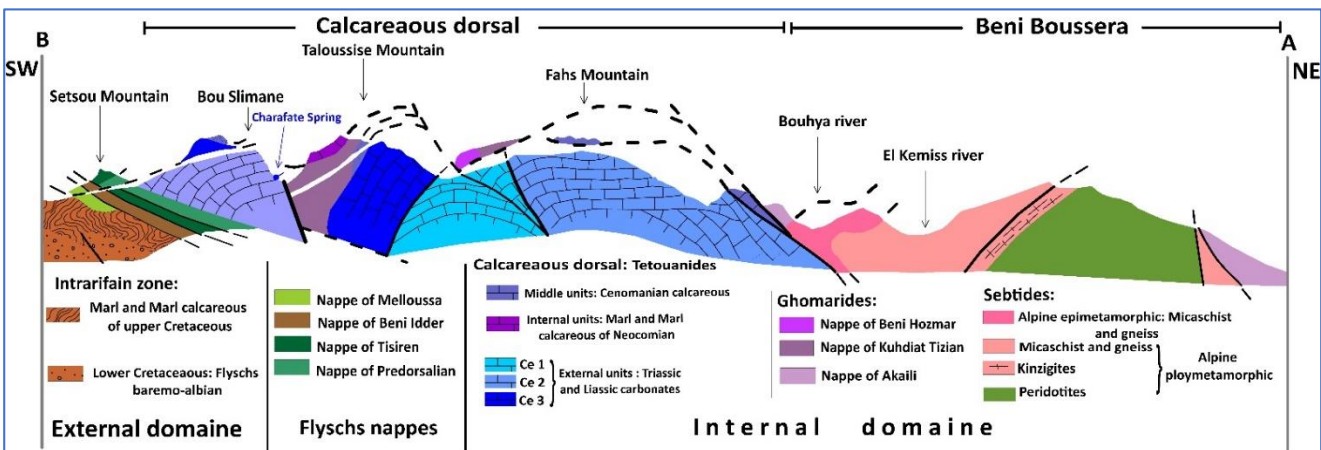

**Figure 4.** Geological cross section (**A**–**B**) along the Calcareous Dorsal (adapted from Geological Rif map 1:500,000).

From a hydrogeological point of view, the Internal Rif region consists of a diversity of geological formations (Figures 3 and 4). Many limited porous media aquifers are located mainly along the downstream of rivers (Martil, Laou, etc.), while the main water resources are provided by mountainous aquifers located mainly in carbonate and fissured formations [31]. The calcareous dorsal is home to one of the few, if not the only, karst of any size worthy of the term in the northern Rif. It is known as an important water resource. This resource, as is known, is in direct causal relationship with the lithology of the Dorsal, the style and geometry of its internal deformation, and with its overall structural layout and the nature of its deep bedrock, to name only a few intrinsic parameters.

The main karst system in the region is the called "Calcareous Dorsal" which can be considered as a single hydrogeological unit due to the important thickness of limestone and dolomite formations and their location above impervious formations. Several springs come out at the base of the limestone ridge following several faults with varying directions (N-S, E-W, NE-SW, and NW-SE), while others are dispersed geographically in the different formations (Figure 3). The most important springs are as follows: Ras El Maa Spring (Sample number 3) located at the west of the Jbel Tissouka unit where it rises above in the flint limestone; Aayaden Spring (Sample number 1) welling in massive carbonate formations of Hettangian age; and Chrafate Spring (Sample number 21), which is part of the Bou Slimane unit and gushes in the flint limestone of Pliensbachian age. For the entire area, these springs are the main water resources for both drinking water and irrigation (Table 1).

### 2.2. Methods and Analytical Techniques

Sixty-seven springs were sampled in various parts of the Calcareous Dorsal of Rif belt, and they were analyzed for their chemical and isotopic compositions (Figure 3). Temperature, pH, electrical conductivity (EC), and total alkalinity were measured in the field. All water samples were shipped to National Center of Energy, Sciences and Nuclear Techniques (CNESTEN) in Morocco for chemical and isotopic measurements. Major elements were

determined by ion chromatography (IC) on a Thermo-Fisher DIONEX-DX120. Bicarbonate ($HCO_3^-$) was determined by using a titration method in the field.

**Table 1.** General characteristics of main Dorsal springs.

| Spring Code | Name | Group | Spring Elevation (m) | Flow Rate | Uses | Contact | Nature of Discharge |
|---|---|---|---|---|---|---|---|
| 2 | Ras El Maa | Lakrâa Mountain | 700 | >100 L·s$^{-1}$ | Drinking water and irrigation (DW and I) | Faults oriented NE-SW and NW-SE | Highly variable |
| 21 | Chrafate | Lakrâa Mountain | 960 | High flow | DW and I | Faults oriented NW-SE, NE-SW and E-W | Highly variable |
| 15 | Onsar Rahmanio | Lakrâa Mountain | 760 | >10 L·s$^{-1}$ | DW and I | Fault and stratigraphic | Variable |
| 26 | Onsar El Fouara | Lakrâa Mountain | 660 | ~5 L·s$^{-1}$ | DW and I | Stratigraphic contact | Constant |
| 17 | Aghbalou | North of Lao River | 840 | >30 L·s$^{-1}$ | DW and I | Fault and stratigraphic | Variable |
| 42 | El Hamma | North of Lao River | 380 | >50 L·s$^{-1}$ | DW and I | Fault and stratigraphic | Highly variable |
| 60 | Khezanat Melhia | North of Lao River | 300 | ~50 L·s$^{-1}$ | DW and I | Lithological and tectonic contact | Constant |
| 5 | Aïn Bousmlal | North of Lao River | 300 | >10 L·s$^{-1}$ | DW and I | Stratigraphic | Variable |
| 57 | Aïn Jamaâ | Haouz end Dersa Mountain | 1300 | ~1 L·s$^{-1}$ | DW and I | Lithological contact | Constant |

Stable isotopes of oxygen and hydrogen were performed by mass spectrometry (Finnigan. Delta Plus) at the CNESTEN Laboratory. The results for stable isotopes are expressed in conventional notation versus V-SMOW, with analytical uncertainty of 0.08‰ for $\delta^{18}O$ and 0.9‰ $\delta^2H$.

Tritium concentrations were analyzed at CNESTEN laboratory by using the liquid scintillation counting method after electrolytic enrichment. Tritium contents are expressed in Tritium Units (1 T.U = 3.24 pCi/L; 1 atom of Tritium for $10^{18}$ of hydrogen atoms).

The $\delta^{13}C$ and $^{14}C$ determinations were made on the TDIC (Total Dissolved Inorganic Carbon) of groundwater and precipitated in the field as $CaCO_3$ at a pH higher than 9.0. The $^{14}C$ content is measured using a liquid scintillation counter and expressed in pmC (percentage of modern Carbon). The uncertainty associated with this method vary with the amount of carbon available in each sample and increases where $^{14}C$ content is low. The $\delta^{13}C$ values, obtained in the TDIC by mass spectrometry, are reported in ‰ to V-PDB (Vienna-Pee Dee Belemnite) standard, with an uncertainty of $\pm 0.1$‰. The $\delta^{13}C$ and $^{14}C$ analyses were performed in Vienna (Austria) at the laboratories of International Atomic Energy Agency-Isotope Hydrology Section.

In this study, principal component analysis (PCA) was applied to 13 variables (physicochemical and isotopic parameters) of 67 observations (spring's groundwater samples) from Calcareous Dorsal aquifers to produce principal components explaining the different processes that control groundwater hydrochemical origin and variation during recharge and groundwater movement. This multivariable statistical method has been frequently used in hydrogeochemical characterization studies [32,33] in order to facilitate the interpretation of results.

## 3. Results and Discussion

The statistical results (minimum, maximum, mean, and standard deviation) of chemical and isotopic composition of samples collected from springs were presented in Table 2.

### 3.1. Groundwater Hydrochemistry Processes

The in situ measured pH shows that the groundwater samples are of slightly alkaline to alkaline types. The pH values ranged from 7.06 to 8.80 with an average of 7.79 (Table 2). Electrical conductivity (EC) varied from 243 to 1130 µS/cm, which shows that the groundwater samples of studied area are freshwater.

**Table 2.** Descriptive statistics of physico-chemical and isotopic parameters of karstic springs in Calcareous Dorsal of Rif. EC: Electrical Conductivity; D: Deuterium.

| Variable | Observations | Minimum | Maximum | Mean Value | Std. Deviation |
|---|---|---|---|---|---|
| T (°C) | 67 | 9.00 | 26.00 | 16.44 | 3.09 |
| pH | 67 | 7.06 | 8.80 | 7.79 | 0.36 |
| EC (µS/cm) | 67 | 243.00 | 1130.00 | 541.80 | 178.99 |
| $HCO_3^-$ (mg/L) | 67 | 118.80 | 1024.80 | 553.81 | 204.76 |
| $Ca^{2+}$ (mg/L) | 67 | 88.17 | 216.43 | 153.78 | 26.64 |
| $Mg^{2+}$ (mg/L) | 67 | 4.86 | 188.57 | 50.76 | 35.09 |
| $Cl^-$ (mg/L) | 67 | 14.18 | 425.40 | 98.67 | 84.12 |
| $Na^+$ (mg/L) | 67 | 8.24 | 230.95 | 54.00 | 45.56 |
| $NO_3^-$ (mg/L) | 67 | 0.00 | 26.90 | 3.55 | 4.43 |
| $\delta^{18}O$ (‰ vsVSMOW) | 67 | −7.50 | −4.78 | −6.14 | 0.88 |
| $\delta^2H$ (‰ vs VSMOW) | 67 | −43.75 | −22.36 | −32.76 | 5.96 |
| D excess (‰) | 67 | 6.41 | 24.78 | 16.41 | 3.17 |
| $^3H$ (TU) | 30 | 1.90 | 6.68 | 3.76 | 0.98 |
| $\delta^{13}C$ (‰ PDB) | 10 | −14.39 | −9.69 | −12.72 | 1.52 |
| $^{14}C$ (pmC) | 10 | 79 | 98 | 89.5 | 6.45 |

The $HCO_3^-$ concentrations are between a minimum of 118.8 mg/L and a maximum of 1024.8 mg/L, with a mean value of 553.81 mg/L. This concentration of $HCO_3^-$ can be explained by carbonate dissolution.

The low concentration for other chemical elements ($Ca^{2+}$, $Mg^{2+}$, and $Cl^-$) is observed. The average concentrations of cations and anions followed the order of $Ca^{2+} > Na^+ > Mg^{2+}$ and $HCO_3^- > Cl^- > NO_3$, respectively. Herein, $Ca^{2+}$ and $HCO_3^-$ were the dominated cation and anion in the groundwater system of the study area.

Table 3 presents the correlation matrix produced by PCA and includes the correlation coefficient (r) between 13 variables using Pearson's correlation coefficient. Low to moderate correlation coefficient between salts and EC suggest minor influence of water–rock interactions. The pH shows a negative correlation with almost all major elements and temperature.

**Table 3.** Proximity matrix among various physico-chemical and isotopic parameters using Pearson's correlation coefficient (r).

| | T°C | pH | EC | $HCO_3^-$ | $Ca^{2+}$ | $Mg^{2+}$ | $Cl^-$ | $Na^+$ | $NO_3^-$ | $\delta^{18}O$ | $\delta^2H$ |
|---|---|---|---|---|---|---|---|---|---|---|---|
| **T°C** | 1 | | | | | | | | | | |
| **pH** | −0.02 | 1 | | | | | | | | | |
| **EC** | 0.53 | −0.56 | 1 | | | | | | | | |
| **$HCO_3^-$** | 0.27 | −0.33 | 0.61 | 1 | | | | | | | |
| **$Ca^{2+}$** | 0.13 | −0.55 | 0.47 | 0.23 | 1 | | | | | | |
| **$Mg^{2+}$** | 0.12 | −0.03 | 0.34 | 0.38 | −0.14 | 1 | | | | | |
| **$Cl^-$** | 0.35 | 0.18 | 0.21 | 0.21 | 0.01 | 0.42 | 1 | | | | |
| **$Na^+$** | 0.35 | 0.18 | 0.21 | 0.21 | 0.01 | 0.42 | 0.98 | 1 | | | |
| **$NO_3^-$** | 0.23 | −0.05 | 0.18 | −0.03 | 0.05 | 0.13 | 0.26 | 0.26 | 1 | | |
| **$\delta^{18}O$** | 0.44 | −0.38 | 0.53 | 0.42 | 0.27 | −0.09 | −0.01 | −0.01 | 0.08 | 1 | |
| **$\delta^2H$** | 0.40 | −0.32 | 0.45 | 0.47 | 0.18 | −0.01 | 0.07 | 0.07 | 0.02 | 0.90 | 1 |

The low values of $NO_3$ (0.0 to 26.9 mg/L) and the non-existent correlation with $\delta^{18}O$ (r = 0.08), $^2H$ (r = 0.02), and EC (0.18), indicating the minimal effect of anthropogenic activity in the recharge area. All samples show values lower as the WHO water quality guidelines of nitrates (<50 mg/L). In the recharge area of the studied aquifer, plant cover is generally dense, which is the main source of nitrate intake when the level of groundwater

is below 10 mg/L. When nitrate level in groundwater exceeds 50 mg/L, the excessive use of agriculture fertilizers and domestic and industrial effluents are the main sources of nitrate contamination.

According to PCA treatment, the largest three eigenvalues are 34.40, 24.31, and 12.33% of the variance (Table 4). PCA results shows that the first and second principal components (eigenvalues) explain 58.71% of the total variance of variables, and three main eigenvalues summarized 71.04% of the total information described by the nine groundwater variables used in the study. The two main principal components (58.71% of the variance) were used to explain the hydrochemical processes of the study area.

**Table 4.** Principal component analysis of groundwater hydrochemical parameters.

|  | F1 | F2 | F3 | F4 | F5 |
|---|---|---|---|---|---|
| T | 0.62 | 0.01 | 0.31 | −0.16 | 0.65 |
| pH | −0.29 | 0.80 | −0.01 | −0.14 | 0.27 |
| EC | 0.77 | −0.50 | −0.03 | 0.06 | 0.18 |
| $HCO_3^-$ | 0.63 | −0.33 | −0.47 | −0.01 | 0.14 |
| $Ca^{2+}$ | 0.33 | −0.65 | 0.33 | −0.32 | −0.35 |
| $Mg^{2+}$ | 0.58 | 0.24 | −0.54 | 0.40 | −0.11 |
| $Cl^-$ | 0.73 | 0.58 | 0.07 | −0.23 | −0.24 |
| $Na^+$ | 0.73 | 0.58 | 0.07 | −0.23 | −0.24 |
| $NO_3^-$ | 0.36 | 0.15 | 0.61 | 0.65 | −0.06 |
| Eigenvalue | 3.09 | 2.18 | 1.11 | 0.84 | 0.81 |
| Variability (%) | 34.40 | 24.3 | 12.33 | 9.38 | 9.09 |
| Cumulative % | 34.40 | 58.7 | 71.04 | 80.42 | 89.52 |

The loadings values of groundwater variables T, EC, $HCO_3$, $Mg^{2+}$, $Cl^-$, and $Na^+$ were calculated to be 0.62, 0.77, 0.63. 0.58, 0.73, and 0.73, respectively. PC1 had a high contribution of loading factors for the above parameters in PCA results. PC1 demonstrated the geogenic sources in the study area by showing a high contribution of moderate and strong positive loadings for all groundwater samples. PC1 indicates the ionic configuration of groundwater resultant especially from hosted rocks dissolution.

For PC2, the high loadings values of groundwater were pH (r = 0.80) and $Ca^{2+}$ (r = −0.65. However, $NO_3$ is correlated with PC3 and PC4 (0.61 and 0.65, respectively) with a low eigenvalue of (1.11 and 0.84, respectively), which explain the absence of anthropogenic sources in the study area.

PCA results (Figure 5) revealed that the geogenic source (i.e., dissolution of carbonates) is a dominant hydrogeochemical process.

*3.2. Stable Hydrogen and Oxygen Isotopes*

Descriptive statistics of stable isotopes of water molecule show that $\delta^{18}O$ and $\delta^2H$ in groundwater from the Calcareous Dorsal springs ranged from −7.5 to −4.8‰ and −44 to −22‰, respectively (Table 2). The mean values were −6.1 and −33‰ for $\delta^{18}O$ and $\delta^2H$, respectively. The values of $\delta^{18}O$ and $\delta^2H$ for all sampled springs are projected in a scatter-plot diagram to show their relationship, and they were compared to the Global Meteoric Water Line (GMWL), the Local Meteoric Water Line (LMWL), the Eastern Mediterranean Sea Meteoric Water Line (NIR), and Gibraltar Meteoric Water Line (GiMWL) elaborated from Gibraltar station of Global Network of Isotopes in Precipitation (GNIP).

Figure 6 confirms that all springs cluster into three groups. The first group (Haouz and Dersa Mountain) has enriched values of $\delta^{18}O$ and $\delta^2H$ and ranged from −5.4 to −4.8‰ and −28 to −22‰, respectively. This group plot close to both LMWL and GiMWL fits rather well with the long-term mean values of Gibraltar GNIP station. This group is characterized by precipitations associated with humid air masses coming from the Atlantic Ocean.

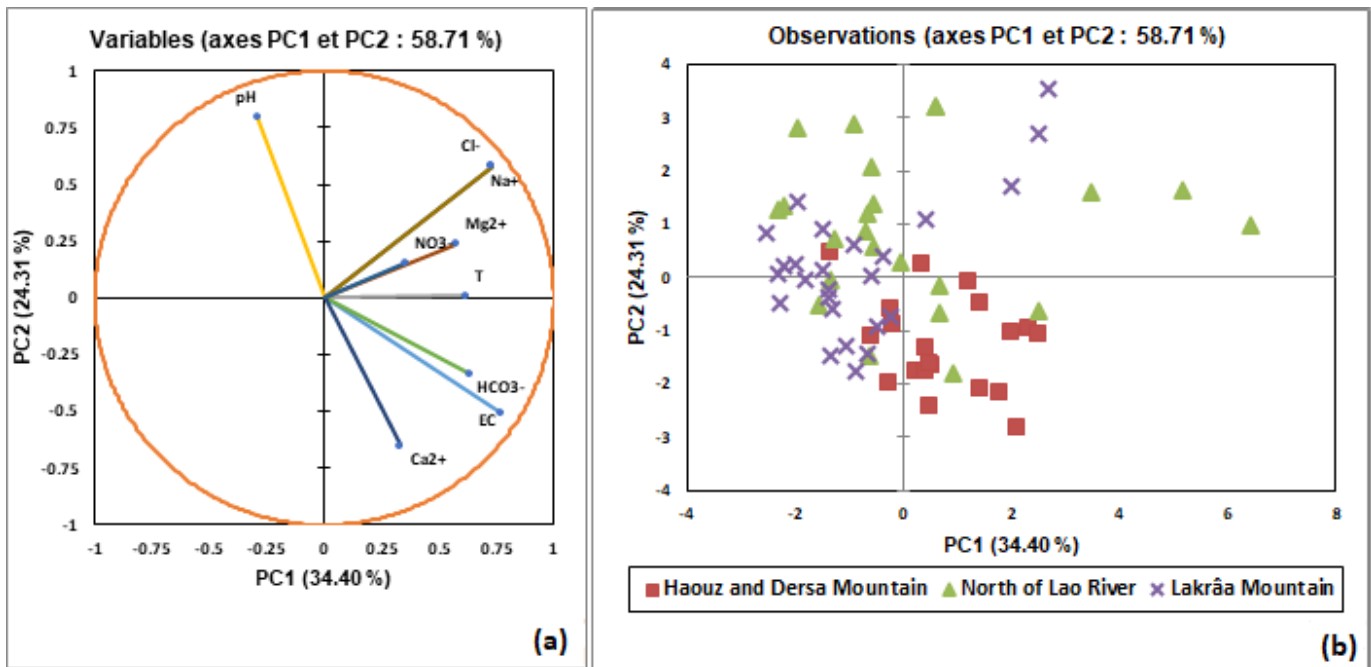

**Figure 5.** Plot of principal component 1 (PC1 = 34.40%) versus principal component 2 (PC2 = 24.31%) for principal component analysis of hydrochemical data in the study area. Factor loadings of 9 variables (**a**) and factor scores of 67 samples (**b**) of PC1 and PC2 (58.71%).

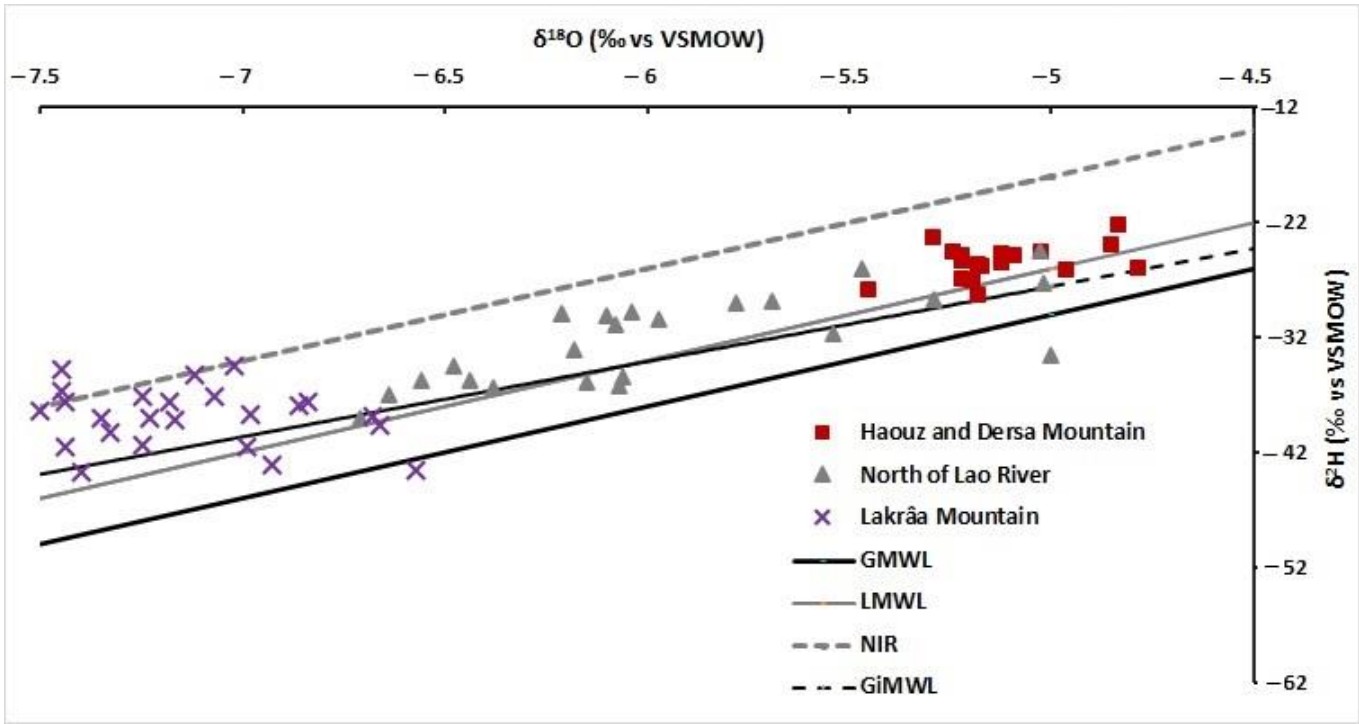

**Figure 6.** $\delta^2H$ versus $\delta^{18}O$ values of groundwater as compared to the Global Meteoric Water Line (GMWL: $\delta^2H = 8\ \delta^{18}O + 10$ [35]), Local Meteoric Water Line (LMWL: $\delta^2H = 8\ \delta^{18}O + 14$ [36], Eastern Mediterranean Sea Meteoric Water Line (NIR: $\delta^2H = 8\ \delta^{18}O + 22$ [34]) and to Gibraltar Meteoric Water Line (GiMWL).

The second group (Lakrâa Mountain), with depleted stable isotope ratios ($\delta^{18}O$ and $\delta^2H$ ranged from $-7.5$ to $-6.6$‰ and $-43$ to $-35$‰, respectively), shows a trend towards

the NIR line defined for the East Mediterranean Sea [34]. This trend suggests that the aquifer is recharged in this part by water derived from rainfall and from East Mediterranean Sea air masses.

The third group (North of Lao River), with intermediate stable isotope ratios between the first and the second group, has been influenced to varying degrees by precipitation from the Atlantic and eastern Mediterranean Sea.

In summary, the majority of the samples plotted in the region between GMWL and NIR lines show that precipitating air masses were coming mainly from the North Atlantic and travelled over the western part of the Mediterranean Sea, where it interacted more strongly with cold air masses.

Figure 7 indicates that $\delta^{18}O$ and deuterium excess (d) have a significant inverse connection. The negative relationship between d and $\delta^{18}O$ in spring water demonstrates that when d decreases, $\delta^{18}O$ levels steadily increase. This, supports the hypothesis that evaporation is the secondary factor influencing oxygen and hydrogen isotopic fingerprints during the rainfall events in the Mediterranean environment. It also implies that present groundwater is mixed with water from various rainfall episodes with varying $\delta^2H$ and $\delta^{18}O$ values. The increase in d from groundwater values to the rainfall values indicates that groundwater and rainwater have mixed.

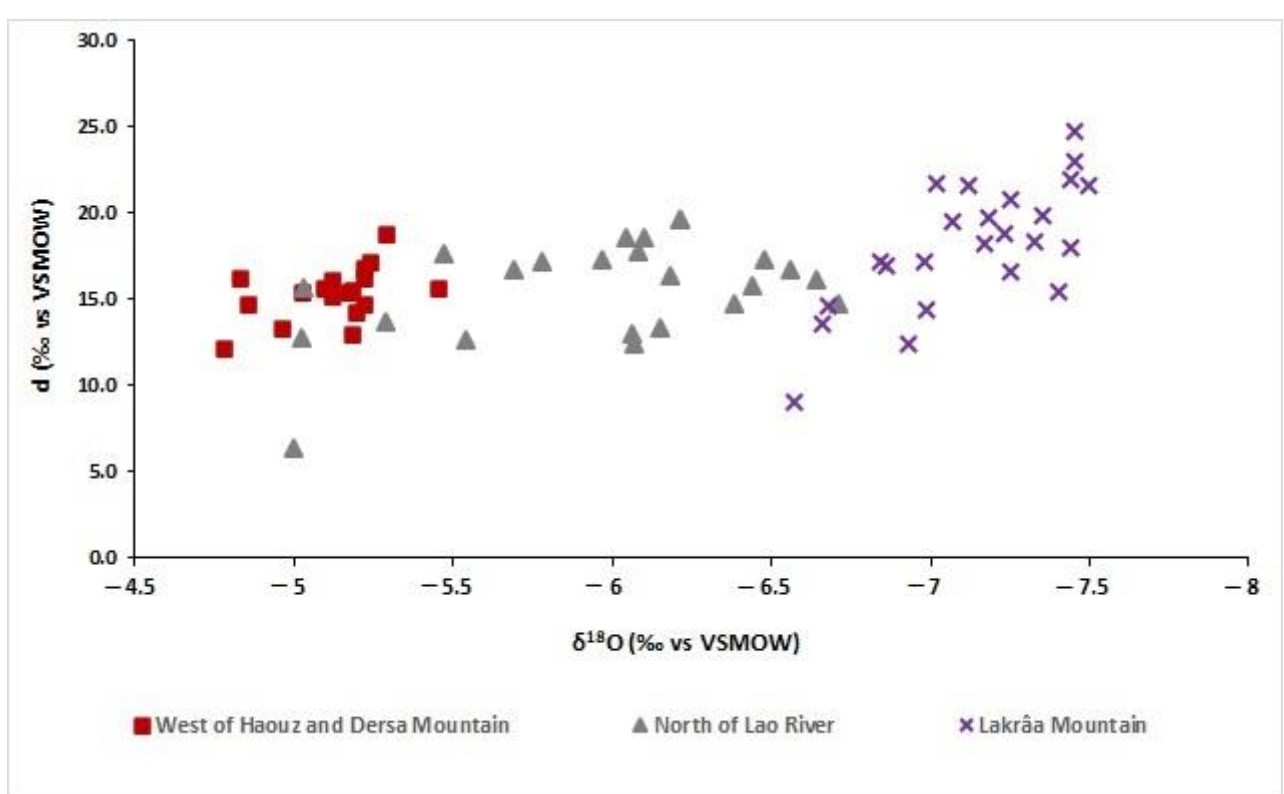

**Figure 7.** Reverse relationship between deuterium excess (d in ‰ vs. VSMOW) and δ18O (‰ vs. SMOW) values in groundwater samples from Calcareous Dorsal springs.

High d-excess values in the springs of Lakraâ Mountain often indicate increased moisture recycling in this area, which may be impacted by humid air masses from the Mediterranean Sea. Low values can be found north of the Lao River as well as north of Haouz and Dersa Mountain.

### 3.3. Identifying Recharge Processes

The mean recharge elevation for Calcareous Dorsal springs is determined by using standard oxygen and hydrogen isotope interpretations in this study.

Several studies have used rainwater surveys in Morocco to determine the altitudinal gradient for $\delta^{18}O$. The early ones mention a $-0.3$ in $\delta^{18}O$ per 100 m altitudinal gradient [37]. For $\delta^{18}O$ per 100 m, $-0.27$ was discovered by calculating the regional oxygen gradient utilizing the relationship between altitude emergence points of some examined springs and their $\delta^{18}O$ contents [31]. However, Stitou El Massari et al. [38] observed $-0.33$ $\delta^{18}O$ per 100 m in northern Morocco.

Ait Brahim et al. [39], using spatiotemporal data from GNIP and rain gauged stations, established a contrasted altitudinal gradient of 0.11 to 0.18‰ per 100 m. This contrast depends on the transect across Morocco. In this study, the altitudinal gradient of 0.18‰ per 100 m is adopted, because it corresponds to established altitudinal gradient along the NE-SW transect across Morocco [39]. Using this oxygen gradient, the relationship between altitude emergence of sampled springs and their $\delta^{18}O$ contents is established (Figure 8).

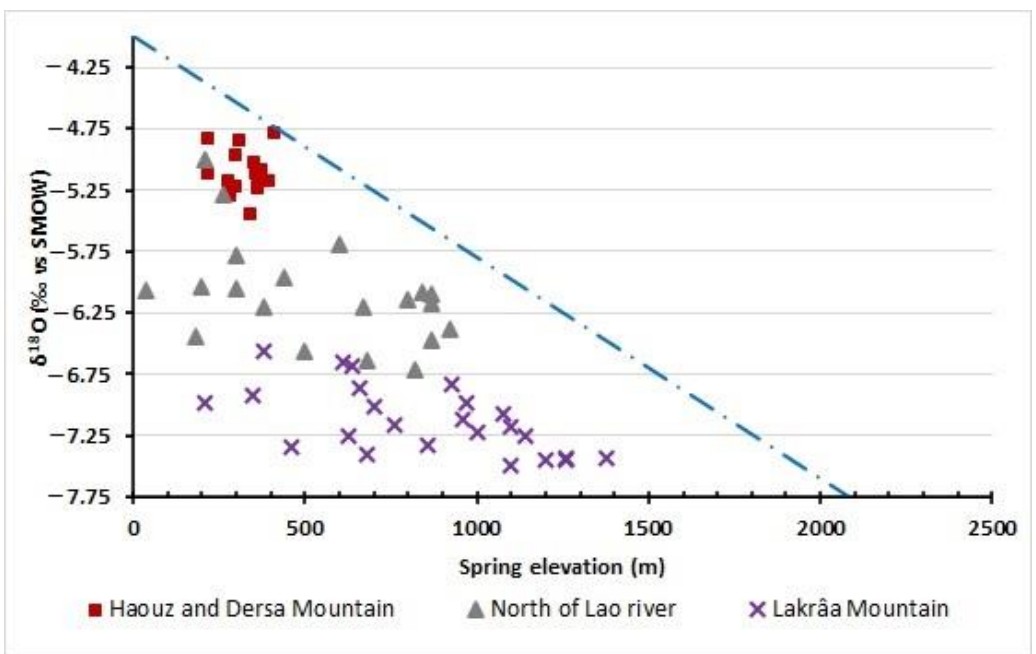

**Figure 8.** Determination of groundwater recharge elevation for groundwater discharging from springs. The broken blue line defines the relationship between $\delta^{18}O$ composition of precipitation, and elevation ($^{18}O = 0.0018 \times Zm-4$) was determined from an analysis of rainwater samples by [39]. Different symbols show spring elevation and $\delta^{18}O$ compositions of spring discharge.

The altitude of the recharge area of the first group (Haouz and Dersa Mountain) varies between 400 and 800 m.a.s.l. The sampled springs from the second group (Lakrâa Mountain) recharged at altitudes ranging from 1400 to 2000 m.a.s.l., which is quite close to the highest summits of neighboring mountains (e.g., Lakrâa Mountain, which has highest summit of roughly 2159 m.a.s.l.). The intermediate group (north of Lao river) obtains its water from an altitude ranging between 550 and 1500 m.a.s.l. (Figure 8).

### 3.4. Radiocarbon ($^{14}C$) and Stable Isotopes of Carbon ($\delta^{13}C$)

Table 5 shows the dissolved inorganic carbon (DIC), mostly in the form of $HCO_3^-$, radiocarbon activity ($^{14}C$), and $\delta^{13}C$ values of DIC for 10 water samples from springs.

In comparison to VPDB, the $\delta^{13}C$ values range from $-9.7$ to $-14.4$ with an average of $-12.72$. The $^{14}C$ activities range from 79 to 78 pmC, with an average of 89.5 pmC (Table 5).

Differences in aquifer properties, primarily soil $CO_2$, can explain the relatively small variance in isotopic compositions. As a result, the measured average value of $\delta^{13}C$ DIC ($-12.72$) corresponds to the $CO_2$ composition produced by soil respiration and carbonate

dissolution. The $\delta^{13}$C values of such plants (Calvin, C3, photosynthetic cycle) would be close to $-27$ V-PDB based on the facts reported in Section 2.1 and Clark and Fritz [40].

**Table 5.** 3H, (DIC), $^{14}$C, $\delta^{13}$C, and radiocarbon age (IAEA Model) of springs groundwater samples from Calcareous Dorsal of Rif Mountains.

| Group | Sample/Code | $^3$H/TU | (DIC)/(mg/L) | $\delta^{13}$C/‰ | $^{14}$C/PmC | ± | s | q | Corrected Age/Years | Uncorrected Age/Years |
|---|---|---|---|---|---|---|---|---|---|---|
| **Haouz and Dersa Mountain** | 3 | 3.96 ± 0.1 | 610.0 | −12.5 | 86.0 | ± | 1.0 | 0.92 | 644 | 1247 |
| | 7 | 4.01 ± 0.1 | 854.0 | −14.4 | 87.0 | ± | 0.3 | 1.02 | 460 | 1151 |
| | 4 | NA | 610.0 | −13.4 | 89.0 | ± | 0.3 | 0.97 | 823 | 963 |
| **North of Oued Lao** | 5 | 4.53 ± 0.1 | 585.6 | −14.0 | 88.0 | ± | 0.4 | 1.00 | 183 | 1057 |
| | 9 | 2.72 ± 0.1 | 390.0 | −13.2 | 95.0 | ± | 0.8 | 0.95 | 164 | 424 |
| | 25 | 3.41 ± 0.1 | 378.2 | −9.7 | 82.0 | ± | 0.3 | 0.76 | 488 | 1641 |
| | 59′ | 3.23 ± 0.1 | 663.6 | −11.0 | 98.0 | ± | 0.4 | 0.83 | 224 | 167 |
| **Lakrâa Mountain** | 21 | 4.20 ± 0.1 | 317.2 | −11.7 | 79.0 | ± | 0.4 | 0.87 | 939 | 1949 |
| | 32 | 2.87 ± 0.1 | 732.0 | −13.4 | 93.0 | ± | 0.3 | 0.97 | 458 | 600 |
| | 66 | 3.39 ± 0.1 | 610.0 | −14.1 | 98.0 | ± | 0.4 | 1.01 | 339 | 1167 |

The dominant geochemical processes affecting the $\delta^{13}$C and $^{14}$C contents of DIC (e.g., geochemical reactions, carbon isotopic exchange, $^{14}$C decay, and mixing of water) and which models are most appropriate for radiocarbon dating of groundwater system of Calcareous Dorsal can be recognized by using Han and Plummer's graphical method [41–43].

The resultant graphs are shown in Figure 9. Graph (a) represents the relation between $\delta^{13}$C and reciprocal of dissolved inorganic carbon concentration (1/[DIC]), graph (b) shows the relation between $^{14}$C and 1/[DIC], and finally the graph (c) shows the relationship between $^{14}$C and $\delta^{13}$C.

According to graphical analysis, the complete transformation of $CO_2$ rich in $^{14}$C into $HCO_3^-$ is the dominant geochemical processes affecting the $\delta^{13}$C and $^{14}$C content of DIC. In this case, the high value of $^{14}$C can be explained by the quick movement of water into the groundwater system.

The same hypothesis is also confirmed by $\delta^{13}$C values that are plotted between $\delta^{13}$Ca2 and $\delta^{13}$Ci. These values could be explained primarily by carbon exchange between soil $CO_2$(g) and HCO3- in water.

The relatively low $\delta^{13}$C values show that dissolution of soil $CO_2$ with minor carbon exchange between DIC and carbonate minerals are the dominant processes affecting the $\delta^{13}$C and $^{14}$C content of DIC in the Calcareous Dorsal groundwater.

Additionally, the enrichment of alkalinity in groundwater is an indicator of the consumption of carbon dioxide in mineral weathering.

*3.5. Groundwater Residence Time*

3.5.1. Tritium Content

Figure 10 shows the results of the comparison between tritium contents measured in groundwater from the springs of Calcareous Dorsal with tritium contents in precipitation in the nearest GNIP stations; GNIP of Gibraltar since 1955s until 2017 and GNIP of Fez-Saiss. Tritium in the atmosphere peaked in the Gibraltar GNIP station in 1963 at 1100 TU and then declined slowly, as observed in the same station (Figure 10. Actually, the mean value of tritium in precipitation in Mediterranean area (Gibraltar GNIP station) is measured to be about 2.5 TU.

This comparison suggests that the tritium values from groundwater of springs are similar to contemporary rains from both Gibraltar and Fes-Saiss stations situated around 100 km north and 250 km south of the study area, respectively. This evidence indicates that groundwater in the Calcareous Dorsal was recharged by young water based on high contents of tritium in groundwater (up to 5.8 TU), i.e., contains a component younger than 60 years.

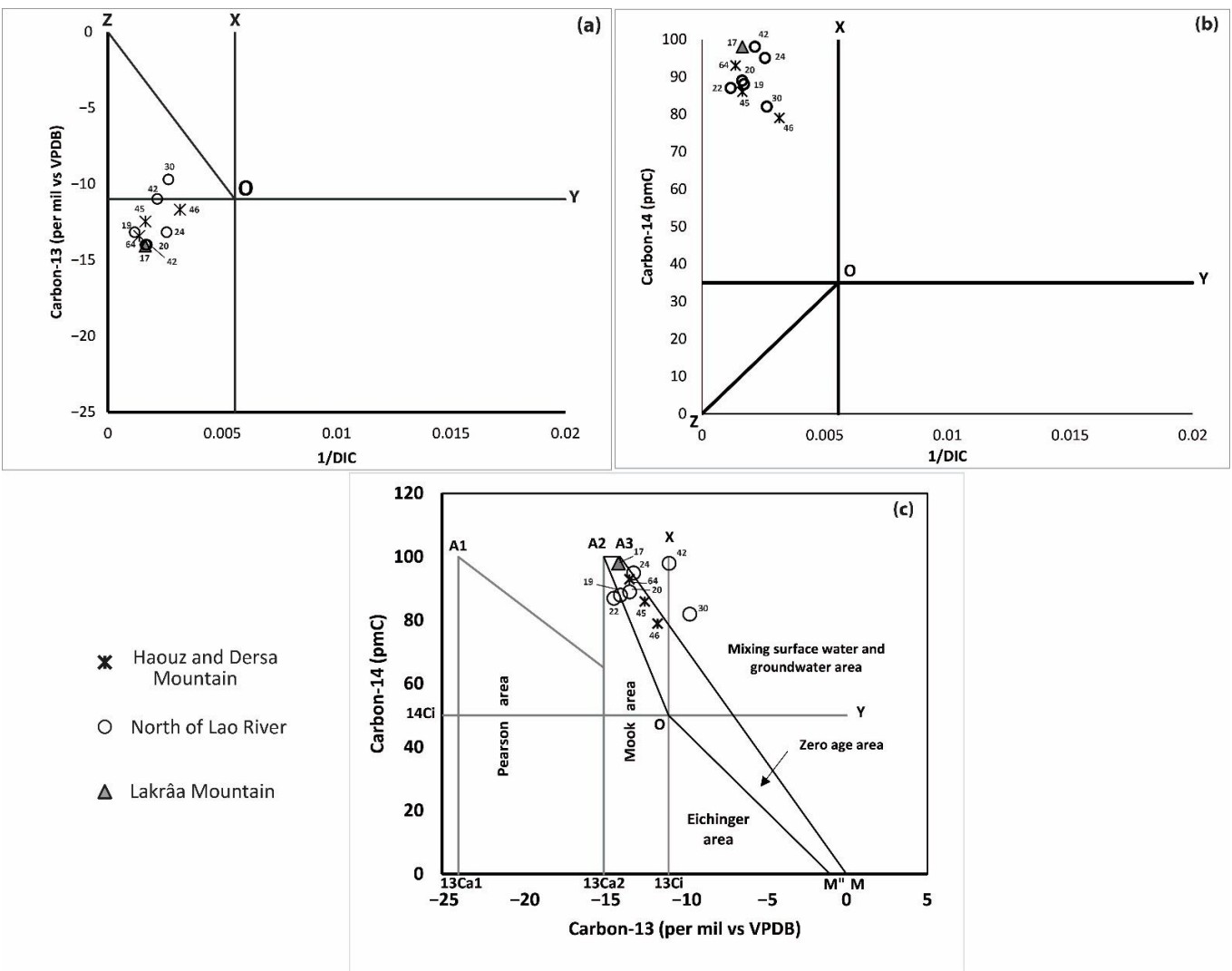

**Figure 9.** Relation among [14]C activity, carbon isotopic composition, and DIC in groundwater collected for this study; the use of the Han and Plummer approach [44]. (**a**) Relation between [14]C and 1/[DIC]; (**b**) Relation between $\delta^{13}C$ and 1/[DIC]; and (**c**) Relation between $\delta^{13}C$ and [14]C activity. Based on Han and Plummer [41], point A1 represents carbon isotopic composition of dissolved soil $CO_2$ in equilibrium with soil gas $CO_2$ ($\delta$ [13]Ca1 = $-24$‰ and [14]Ca1 = 100 pmC), A2 or Mook's point represents $HCO_3$- equilibrated with soil $CO_2$ ($\delta$ [13]Ca2 = $-14$‰ and [14]Ca2 = 100 pmC), A3 represents a mixture of $CO_2$(aq), and $HCO_3$- equilibrated with soil $CO_2$, point O or Tamers' point ([14]Ci = 0.5 [14]Ca1, $\delta^{13}$Ci = 0.5 $\delta$ [13]Ca1), represents the initial carbon isotopic composition of the DIC after isotope exchange; M represents the carbonate rock ($\delta^{13}Cs$ = 0‰ and [14]Cs = 0) and M″ or Eichinger's point ($\delta^{13}CE$ = $-1$‰ and [14]CE = 0) represents $\delta^{13}C$ value of DIC ($CO_2$(aq) and $HCO_3$-) in equilibrium with solid carbonate. The O-M″ Line represents the zero-age line in keeping with Eichinger's model, A3-M is the IAEA Line and A2-O-M″ is the Han and Plummer Line.

### 3.5.2. Radiocarbon Dating

In this case study, the $\delta^{13}C$ value ranges between $\delta^{13}$Ca2 and M (Figure 10), and according to Han et al. [41], IAEA's model can be used similarly to Han and Plummer's model. The IAEA model assumes mixed open and closed systems. The IAEA model assumes that the isotopic composition of the DIC evolved initially under completely open-system conditions.

The groundwater residence time of springs from Calcareous Dorsal is determined using IAEA model. The radiocarbon dating method using the IAEA model (Table 5) indicates ages ranging from present to several hundred years, reflecting modern recharge.

All the samples analyzed for radiocarbon have detectable tritium ($^3$H), which is an indicator of recent recharge [40]. Groundwater in recharge zones shows the presence of tritium and uncorrected $^{14}$C ages less than 2000 y BP (Table 5).

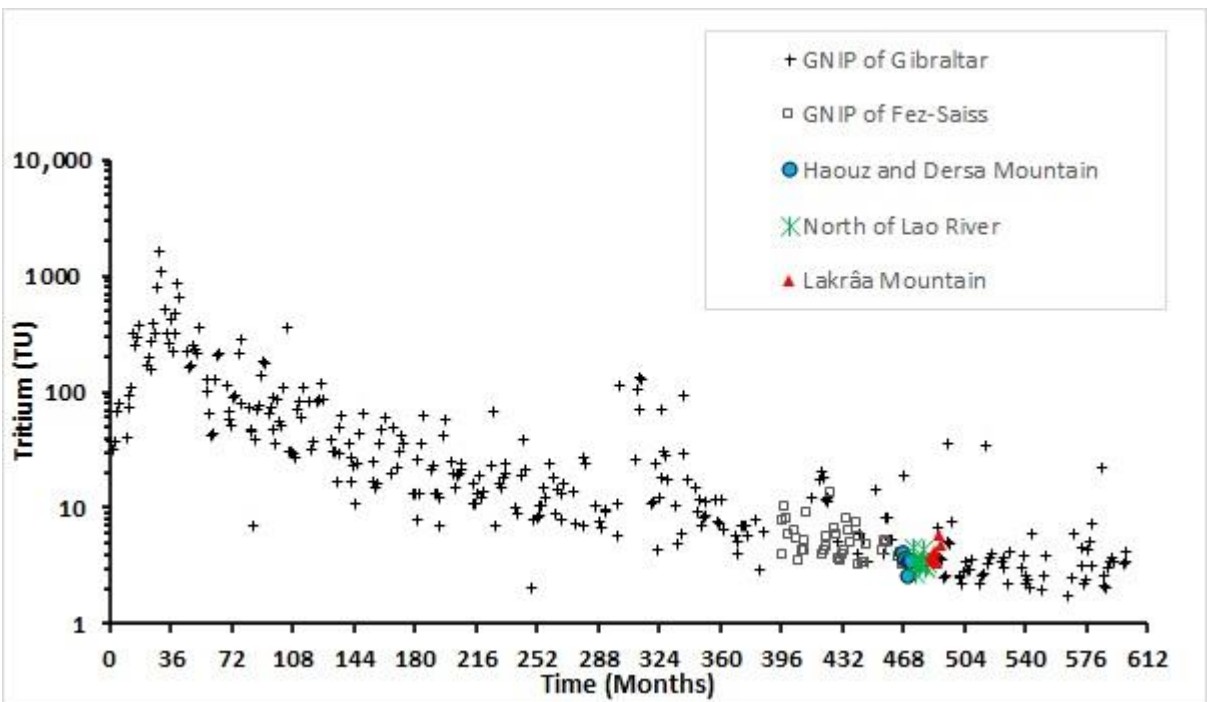

**Figure 10.** Tritium values measured in karstic springs of Calcareous Dorsal compared to GNIP values in Gibraltar and Fez-Saiss stations.

## 4. Conclusions

The results obtained from statistical analysis indicate that cation and anion, expressed in milligram per liter (mg/L), predominance in Calcareous Dorsal springs is in the following order $Ca^{2+} > Na^+ > Mg^{2+}$ and $HCO_3^- > Cl^- > NO_3^-$, respectively. Herein, $Ca^{2+}$ and $HCO_3^-$ were the dominant cation and anion in the groundwater system of the study area, which explain that the geochemical process of groundwater was controlled particularly by carbonates dissolution.

The stable isotopes $\delta^{18}O$ and $\delta^2H$ show that aquifer recharge is ensured by direct infiltration of oceanic precipitation without significant evaporation. The relationships between altitude emergence points of some studied springs and their $\delta^{18}O$ contents were also used to determine the elevation of the recharge area using the established relationship between altitude and $\delta^{18}O$ in precipitation ($\delta^{18}O = -0.18$ per mil per 100 m). The selected springs have estimated recharge elevations, which are very close to the altitudes of the neighboring mountains.

The study of radioactive isotopes ($^3$H and $^{14}$C) shows that the recharge of springs from Calcareous Dorsal is modern with an actual age according to the IAEA model.

Isotopic and chemical tracings indicate that the karstic aquifer of Calcareous Dorsal is vulnerable to global changes. The recharge area with all springs should be protected from any sources of pollution. In addition, the direct relationship between springs and rainfall makes this aquifer very vulnerable to climate change and variability.

The results of this paper can be used by water managers and stakeholders for better management and protection of this vital resource in such a karstic area.

**Author Contributions:** Conceptualization, M.H., M.Q. and L.B.; methodology, M.B., L.B. and M.H.; software, M.H.; resources, M.Q.; data curation, H.M. and M.Q.; writing—original draft preparation, M.H.; writing—review and editing, M.H., M.Q., L.B., M.B. and J.S.E.M.; project administration, L.B. All authors have read and agreed to the published version of the manuscript.

**Funding:** This research received no external funding.

**Institutional Review Board Statement:** Not applicable.

**Informed Consent Statement:** Not applicable.

**Data Availability Statement:** The data presented in this study are available upon request from the corresponding author.

**Acknowledgments:** The authors express their gratitude to the Hydraulic Department of Rabat and Hydraulic Basin Agency of Loukkos for access to their database. This work was carried out in part within the CHARISMA Project with the assistance of the Hassan 2 Academy of Sciences and techniques and the support of Ibn Zohr University and AgriMeed project.

**Conflicts of Interest:** The authors declare no conflict of interest.

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
