# Peer review of "Isotopic and Chemical Tracing for Residence Time and Recharge Mechanisms of Groundwater under Semi-Arid Climate: Case from Rif Mountains (Northern Morocco)"

_geosciences, doi:10.3390/geosciences12020074_

Round 1
Reviewer 1 Report
General comments :
- Although english is not my native language, I feel it has to be improved.
- Considering my review, you have to be aware that I am not completely familiar with carbon isotopes.
Abstract is fine, overall.
- line 29 : “These values” : do authors refer to tritium data ?
Introduction :
- line 52 : some explanations are necessary about “non-conventional waters”.
- line 53 : It is somewhat confusing. If it is an “alternative” source of water, what is the current source ? If the “calcareous dorsal” is the main source, so how the mountain aquifers are an “alternative” ? This needs to be clarified. I guess that maybe they is not a question of “alternative”, but that this ressource is critical and thus the processes need to be better constrained. If it is the case, the authors have to write it clearly.
- line 75 : “sprint race” ? Is it a typo ?
Materials and methods :
- line 86 : please add the reference.
- line 93 : remove “high latitude”.
- The description of the geological and hydrogeological setting is much too long and confusing. The auhors have to better define and focus on the Calcareous Dorsal. Also, all the units defined in the text have to be represented in Figures 1 and 2.
- line 145 : this sentence needs to be rewritten.
- line 149 : there is a typo on reference 38.
- I suppose that groundwaters flow northeastwards. This needs to be clearly stated.
- line 175 : Finnigan.
- line 177 : “precision” does not exist. Replace by “uncertainty”. Same remark for lines 185 and 188.
Results and discussion :
- The authors may want to provide a Piper diagram. This would be useful.
- line 212 : “mean value” instead of “mean” (also Table 1).
- line 227 : upperscript for 3H.
- Considering the stable isotopes of the water molecule, where is the location of the LMWL ? Are the database available ?
- line 240 : add a space.
- line 245 : Gibraltar (typo).
- line 248 : Mountain.
- Deuterium excess : how is it calculated ?
- Considering recharge processes, define m.a.s.l.
- What is the recharge height vs. topography for Haouz and Dersa ?
- line 310 : I don’t understand the Celsius degrees. Is it a typo ?
- line 352 : replace “age” by “residence time”.
- line 358 : “mean value”.
- line 376 : same remark that for line 352.
- 14C residence times are in the order of a thousand years. Do authors consider such “ages” as present-day “ages” ? Otherwise, as they contain tritium, would it be possible by coupling the two chronometers to constrain the possible mixing processes between the “present-day” waters and the “older”, holocene ones ? Would it be possible to derive mixing percentages, and, further, genuine “ages” of “old” holocene waters (lines 383 – 385) ?
Conclusion :
- line 393 : The authors must state that the absence of evaporation processes refer to “atlantic-derived” waters. Otherwise, there is a contradiction with line 267.
Figure 1 et 2 :
- Please utilize the same colours for both. At present it is really confusing. The “external domain” is not represented on Figure 1. Why ? All the structural domains referred in the text should be represented on Figures 1 and 2. The “Calcareous Dorsal” should be delimited on Figure 1.
- Legend of Figure : replace “adopted” by “adapted”.
Figure 4 :
- Please report the present-day datapoints for each of the sites which define the local straight lines.
Figure 6 :
- Why recharge height and spring elevation are not reported on the same figure (double abscissa) ?
- It would be better to have coloured diagrams for all figures.
Table 1 :
- Are detailed data easily available ?
Table 2 :
- EC instead of “C.E.”
Author Response
Responses to reviewer 1:
Thank you for your proficient review. We have revised and updated the manuscript based on your very insightful feedback.
General comments :
- Although english is not my native language, I feel it has to be improved.
- Considering my review, you have to be aware that I am not completely familiar with carbon isotopes.
Abstract is fine, overall.
- line 29 : “These values” : do authors refer to tritium data ?
Yes
Introduction :
- line 52 : some explanations are necessary about “non-conventional waters”.
It’s explained.
- line 53 : It is somewhat confusing. If it is an “alternative” source of water, what is the current source ? If the “calcareous dorsal” is the main source, so how the mountain aquifers are an “alternative” ? This needs to be clarified. I guess that maybe they is not a question of “alternative”, but that this ressource is critical and thus the processes need to be better constrained. If it is the case, the authors have to write it clearly.
Yes we agree with you. We have replaced the “alternative” by “crucial”
- line 75 : “sprint race” ? Is it a typo ?
It’s corrected
Materials and methods :
- line 86 : please add the reference.
The generated error is caused by automated insertion of “Figure 1a”
- line 93 : remove “high latitude”.
Done.
- The description of the geological and hydrogeological setting is much too long and confusing. The auhors have to better define and focus on the Calcareous Dorsal. Also, all the units defined in the text have to be represented in Figures 1 and 2.
This section is updated and only the Internal domain is described.
- line 145 : this sentence needs to be rewritten.
The sentence is rectified.
- line 149 : there is a typo on reference 38.
Its corrected.
- I suppose that groundwaters flow northeastwards. This needs to be clearly stated.
Yes, for some springs but for others the groundwater flow is northwestwards. The groundwater flow of Dorsal springs follows four major directions NW-SE, NE-SW, E-W and N-S.
- line 175 : Finnigan.
Rectified.
- line 177 : “precision” does not exist. Replace by “uncertainty”. Same remark for lines 185 and 188.
Done
Results and discussion :
- line 212 : “mean value” instead of “mean” (also Table 1).
Done
- line 227 : upperscript for 3H.
Rectified.
- Considering the stable isotopes of the water molecule, where is the location of the LMWL ? Are the database available ?
The Local Meteorological Water Line is done using stable isotopes composition of rainwater from rain gauged stations across Morocco and from GNIP station of Fes, Rabat and Gibraltar. Yes, this database is available.
- line 240 : add a space.
Done
- line 245 : Gibraltar (typo).
Done
- line 248 : Mountain.
Done
- Deuterium excess : how is it calculated ?
The d-excess was calculated following this equation d − excess = δ2H − 8 δ18O,
- Considering recharge processes, define m.a.s.l.
Done. M.a.s.l.= Meter above sea level.
- What is the recharge height vs. topography for Haouz and Dersa ?
Generally, there is small differences between recharge area and the emergence altitude of Haouz and Dersa Mountain. This difference can reach a maximum of about 400 m.
- line 310 : I don’t understand the Celsius degrees. Is it a typo ?
Yes, it’s a typo. It’s rectified.
- line 352 : replace “age” by “residence time”.
Done
- line 358 : “mean value”.
Done
- line 376 : same remark that for line 352.
Done
- 14C residence times are in the order of a thousand years. Do authors consider such “ages” as present-day “ages” ? Otherwise, as they contain tritium, would it be possible by coupling the two chronometers to constrain the possible mixing processes between the “present-day” waters and the “older”, holocene ones ? Would it be possible to derive mixing percentages, and, further, genuine “ages” of “old” holocene waters (lines 383 – 385) ?
This difference can be explained by the influence of recent recharge and old water transit.
The objective of Tritium
Conclusion :
- line 393 : The authors must state that the absence of evaporation processes refer to “atlantic-derived” waters. Otherwise, there is a contradiction with line 267.
This contradiction is adressed.
Figure 1 et 2 :
- Please utilize the same colours for both. At present it is really confusing. The “external domain” is not represented on Figure 1. Why ? All the structural domains referred in the text should be represented on Figures 1 and 2. The “Calcareous Dorsal” should be delimited on Figure 1.
The geological setting is updated and only the Internal domain is described. Because of several Faults lines it’s not easy to delimitate the calcareous dorsal in the figure 1.
- Legend of Figure : replace “adopted” by “adapted”.
Figure 4 :
- Please report the present-day datapoints for each of the sites which define the local straight lines.
It’s reported in the cited references.
Figure 6 :
- Why recharge height and spring elevation are not reported on the same figure (double abscissa) ?
The recharge height of sprigs is represented by the discontinued blue line and the X-axis reports the both the recharge height and spring elevation.
- It would be better to have coloured diagrams for all figures.
Done
Table 1 :
- Are detailed data easily available ?
Yes, if it’s necessary we can put it as a supplementary material.
Table 2 :
- EC instead of “C.E.”
Done

Reviewer 2 Report
Review of manuscript titled “ Isotopic and chemical tracing for residence time and recharge mechanisms of groundwater under semi-arid climate: Case from Rif Mountains (Northern Morocco)”
By Mohammed Hssaisoune Lhoussaine Bouchaou, Mohamed Qurtobi, Hamid Marah, Mohamed Beraaouz , and Jamal Stitou El Messari
General comments:
The paper was an interesting case study on the use of naturally occurring isotopic and geochemical tracers from spring discharge to understand the flow dynamics in a karst aquifer in northern Morrocco. It should be considered for publication after some significant revisions. Understandably, some of the language used in the paper sounds quite unusual to a native English speaker (this reviewer). I have highlighted some of the more unusual word choices in yellow in the manuscript and suggest alternative expressions in the comments near the highlighted text. This occurs throughout the manuscript. These alternative word choices are not repeated in the comments that follow, which focus on the technical aspects of the paper.
Detailed comments:
Section 2.2 – Geological and hydrogeological setting: This section provides a rather detailed description of the geology of the study site that is largely irrelevant to the paper. It is dense and jargon filled and appears to have been adapted from the original studies without regard for its necessity or relevance. The authors should edit the section to retain only the details necessary to understand the importance of the study – namely, that it is a complexly folded and thrust-faulted terrain where hydraulic information is limited and karstic carbonate aquifers form the dominant sources of subsurface water. On the other hand, information that would have been very helpful to interpret the geochemical and isotopic tracers is omitted altogether. This information includes climate-related information such as precipitation patterns (annual and seasonal precipitation volumes), storm paths (do storms generally move in from the Atlantic or the Mediterranean?), and a map showing the general topography of the study area so that the reader can assess the topographic influences on stable isotopic compositions. Important hydrogeologic data not present also include general characteristics of the springs themselves: are they associated with faults or stratigraphic contacts that force water to the surface? Are they high-flow, low-flow, highly variable or constant discharge?
Lines 179-181: A check of Clark and Fritz (1997, p.17) indicates that 1 T.U is equivalent to 3.2 pCi/L (not 3.2 pCi/ml) and that 1 T.U. is 1 tritium atom per 1018 hydrogen atoms (not 1018 hydrogen atoms).
Table 2. The variable listed as C.E. in Table 2 should be EC (electrical conductivity) to be consistent with the usage elsewhere in the paper.
Lines 229-232. The PCA analysis presented on lines 229-232 and in Figure 3 requires much more discussion of the methodology and results. Why was this technique employed? Do the two factors (F1 and F2) have a physical significance? Why were ratios (for example Mg/Ca and HCO3/Cl) and derived quantities (for example, deuterium excess) used rather than basic measurements? What is the basis for the conclusion that carbonate dissolution is the dominant geochemical process? (Very likely, but how is this conclusion supported by the PCA).
Lines 266-268. These lines state that the inverse relation between δ18O and deuterium excess (which would be better shown if values on the x-axis increased from left to right) demonstrate that evaporative processes are controlling the “oxygen and hydrogen isotopic fingerprints in the Mediterranean environment.” However, the data plotted against the GiMWL and NIR in Figure 4 don't really show a classic evaporation trend (i.e. isotopic values that plot along a line with a low slope (4 to 5) that diverges from the meteoric water line at a sharp angle). Instead, the data shown in Fig. 4 suggest that different air-masses are recharging the aquifers from north-to south or that progressive "rainout" of the oceanic signal occurs from north to south with distance from the coast and an increase in elevation (again, some indication of topography on an introductory figure would have been helpful.)
Lines 309-311. These lines make several references to degrees Celsius when discussing 14C activities where they should be referring to pmc (percent modern carbon).
Figure 7 caption. Based on the reviewer’s understanding of Fig. 7, the caption makes numerous errors when referring to features in the figure. These errors are best understood by looking at the marked up version of the pdf where the reviewer has highlighted the potentially erroneous terms in yellow and suggested the correct terms.
Figure 8. The agreement between tritium in the spring samples at the time of sampling with the temporal trends in tritium from the IAEA stations suggests residence times of a few months to a few years, contrary to the much longer residence times associated with 14C ages. Groundwater 14C ages in carbonate rocks are difficult to correct and could easily be off by several hundred years, whereas tritium requires no corrections for water/rock interactions. Do the physical settings of the springs allow for such short residence times (local recharge and discharge superimposed on more long-distance flow? This might be determined by whether spring discharge increases dramatically immediately after rainstorms, i.e. the temporal variability of the spring discharge.
Table 3. The purpose of 14C age corrections is to account for dilution of 14C through water/rock interactions so that 14C activities less than 100 pmc are due to radioactive decay alone. Therefore, the corrected ages should always be less than the uncorrected ages, which is not the case for samples 7,5,59 and 66. Either the correction model use does not apply for these samples or the model was applied incorrectly. The authors should investigate and state the cause of this discrepancy.
Lines 392-392 states that the stable isotopes δ18O and δ2H shows that aquifer recharge is ensured by direct infiltration of oceanic precipitation without significant evaporation. I agree with this statement based on Figure 4, which does not show much offset from the local meteoric water lines. However, arguments presented on lines 266-268 in connection with Figure 5 appear to be arguing for evaporation as a significant process. How do you reconcile these two positions?
Line 396. The change in δ18O with elevation on this line should be expressed as Δδ18O = 0.0018 ‰ m-1

Author Response
Responses to Reviewer 2:
General comments:
The paper was an interesting case study on the use of naturally occurring isotopic and geochemical tracers from spring discharge to understand the flow dynamics in a karst aquifer in northern Morrocco. It should be considered for publication after some significant revisions. Understandably, some of the language used in the paper sounds quite unusual to a native English speaker (this reviewer). I have highlighted some of the more unusual word choices in yellow in the manuscript and suggest alternative expressions in the comments near the highlighted text. This occurs throughout the manuscript. These alternative word choices are not repeated in the comments that follow, which focus on the technical aspects of the paper.
Thank you for your professional comments, it’s very much appreciated. As you mentioned on your comments, several problems need to be addressed. All your kind comments are addressed in the revised version
Detailed comments:
Section 1. What are the features of these aquifers that make them especially vulnerable? Can you elaborate?
Karst aquifers are extremely fragile environments and they are affected by specific hazards and impacts such as human activities and climate change.
Section 2.2 – Geological and hydrogeological setting: This section provides a rather detailed description of the geology of the study site that is largely irrelevant to the paper. It is dense and jargon filled and appears to have been adapted from the original studies without regard for its necessity or relevance. The authors should edit the section to retain only the details necessary to understand the importance of the study – namely, that it is a complexly folded and thrust-faulted terrain where hydraulic information is limited and karstic carbonate aquifers form the dominant sources of subsurface water.
This comment was considered. We've updated the way that this section is presented.
On the other hand, information that would have been very helpful to interpret the geochemical and isotopic tracers is omitted altogether. This information includes climate-related information such as precipitation patterns (annual and seasonal precipitation volumes), storm paths (do storms generally move in from the Atlantic or the Mediterranean?), and a map showing the general topography of the study area so that the reader can assess the topographic influences on stable isotopic compositions.
This comment was taken into consideration. We have improved the study area section by adding climate-related information, storm paths and topography (see figure 1 and 2).
Important hydrogeologic data not present also include general characteristics of the springs themselves: are they associated with faults or stratigraphic contacts that force water to the surface? Are they high-flow, low-flow, highly variable or constant discharge?
This comment was taken into consideration. The general characteristics of the springs have been added to study area section.
Lines 179-181: A check of Clark and Fritz (1997, p.17) indicates that 1 T.U is equivalent to 3.2 pCi/L (not 3.2 pCi/ml) and that 1 T.U. is 1 tritium atom per 1018 hydrogen atoms (not 1018 hydrogen atoms).
It’s corrected.
Table 2. The variable listed as C.E. in Table 2 should be EC (electrical conductivity) to be consistent with the usage elsewhere in the paper.
Updated
Lines 229-232. The PCA analysis presented on lines 229-232 and in Figure 3 requires much more discussion of the methodology and results. Why was this technique employed? Do the two factors (F1 and F2) have a physical significance? Why were ratios (for example Mg/Ca and HCO3/Cl) and derived quantities (for example, deuterium excess) used rather than basic measurements? What is the basis for the conclusion that carbonate dissolution is the dominant geochemical process? (Very likely, but how is this conclusion supported by the PCA).
This section is updated and some variables are eliminated even the isotopes. Because the interest of the ACP is to determine the dominant hydrogeochemical. More discussion and results are added to this section.
Lines 266-268. These lines state that the inverse relation between δ18O and deuterium excess (which would be better shown if values on the x-axis increased from left to right) demonstrate that evaporative processes are controlling the “oxygen and hydrogen isotopic fingerprints in the Mediterranean environment.” However, the data plotted against the GiMWL and NIR in Figure 4 don't really show a classic evaporation trend (i.e. isotopic values that plot along a line with a low slope (4 to 5) that diverges from the meteoric water line at a sharp angle). Instead, the data shown in Fig. 4 suggest that different air-masses are recharging the aquifers from north-to south or that progressive "rainout" of the oceanic signal occurs from north to south with distance from the coast and an increase in elevation (again, some indication of topography on an introductory figure would have been helpful.)
Groundwater from the Lakrâa Mountain shows this nonlinearity of the δ18O and δ2H relationship due to the influence of evaporation during rainfall and by Mediterranean humid air masses. Note that secondary evaporation also decreases the deuterium excess and intercept.
The
Lines 309-311. These lines make several references to degrees Celsius when discussing 14C activities where they should be referring to pmc (percent modern carbon).
It’s rectified.
Figure 7 caption. Based on the reviewer’s understanding of Fig. 7, the caption makes numerous errors when referring to features in the figure. These errors are best understood by looking at the marked up version of the pdf where the reviewer has highlighted the potentially erroneous terms in yellow and suggested the correct terms.
All errors are rectified. Thank you for precious help.
Figure 8. The agreement between tritium in the spring samples at the time of sampling with the temporal trends in tritium from the IAEA stations suggests residence times of a few months to a few years, contrary to the much longer residence times associated with 14C ages. Groundwater 14C ages in carbonate rocks are difficult to correct and could easily be off by several hundred years, whereas tritium requires no corrections for water/rock interactions.
Do the physical settings of the springs allow for such short residence times (local recharge and discharge superimposed on more long-distance flow? This might be determined by whether spring discharge increases dramatically immediately after rainstorms, i.e. the temporal variability of the spring discharge.
Yes, the discharge of almost springs is highly variable and sensitive to rainstorms.
Table 3. The purpose of 14C age corrections is to account for dilution of 14C through water/rock interactions so that 14C activities less than 100 pmc are due to radioactive decay alone. Therefore, the corrected ages should always be less than the uncorrected ages, which is not the case for samples 7,5,59 and 66. Either the correction model use does not apply for these samples or the model was applied incorrectly. The authors should investigate and state the cause of this discrepancy.
The corrected age is rectified. It’s a typo issue.
Lines 392-392 states that the stable isotopes δ18O and δ2H shows that aquifer recharge is ensured by direct infiltration of oceanic precipitation without significant evaporation. I agree with this statement based on Figure 4, which does not show much offset from the local meteoric water lines. However, arguments presented on lines 266-268 in connection with Figure 5 appear to be arguing for evaporation as a significant process. How do you reconcile these two positions?
Only for Lakrâa Mountain which is especially influenced by Mediterranean humid air masses and secondary evaporation during rainfall events.
Line 396. The change in δ18O with elevation on this line should be expressed as Δδ18O = 0.0018 ‰ m-1
It’s rectified.
